# Thermomagnetic-Responsive Self-Folding Microgrippers for Improving Minimally Invasive Surgical Techniques and Biopsies

**DOI:** 10.3390/molecules27165196

**Published:** 2022-08-15

**Authors:** Caleigh R. Dunn, Bruce P. Lee, Rupak M. Rajachar

**Affiliations:** 1Department of Biomedical Engineering, Michigan Technological University, Houghton, MI 49931, USA; 2Marine Ecology and Telemetry Research (MarEcoTel), Seabeck, WA 98380, USA

**Keywords:** smart materials, microgrippers, stimuli responsive, soft robotics

## Abstract

Traditional open surgery complications are typically due to trauma caused by accessing the procedural site rather than the procedure itself. Minimally invasive surgery allows for fewer complications as microdevices operate through small incisions or natural orifices. However, current minimally invasive tools typically have restricted maneuverability, accessibility, and positional control of microdevices. Thermomagnetic-responsive microgrippers are microscopic multi-fingered devices that respond to temperature changes due to the presence of thermal-responsive polymers. Polymeric devices, made of poly(N-isopropylacrylamide-co-acrylic acid) (pNIPAM-AAc) and polypropylene fumarate (PPF), self-fold due to swelling and contracting of the hydrogel layer. In comparison, soft metallic devices feature a pre-stressed metal bilayer and polymer hinges that soften with increased temperature. Both types of microdevices can self-actuate when exposed to the elevated temperature of a cancerous tumor region, allowing for direct targeting for biopsies. Microgrippers can also be doped to become magnetically responsive, allowing for direction without tethers and the retrieval of microdevices containing excised tissue. The smaller size of stimuli-responsive microgrippers allows for their movement through hard-to-reach areas within the body and the successful extraction of intact cells, RNA and DNA. This review discusses the mechanisms of thermal- and magnetic-responsive microdevices and recent advances in microgripper technology to improve minimally invasive surgical techniques.

## 1. Introduction

The prevalence of minimally invasive surgery (MIS) in the clinical environment has been quickly growing. With techniques, such as laparoscopic surgery, endoscopic biopsies and miniaturized tools, surgeons can now perform a wide variety of surgical procedures without directly touching or looking at the regions under operation. The paradigm shift from traditional open surgery to minimally invasive surgery has offered several advantages to patients and health care workers. During open surgery, most of the pain and trauma afflicted on the patient is the result of gaining access to the procedural site rather than the procedure itself [1]. 

Large incisions are often required for the surgeons to directly access the organs under operation. However, in minimally invasive surgery, surgeons can operate through tiny incisions or natural orifices of the body utilizing scopes and micro-scaled tools. This technology has drastically reduced patients’ recovery time as MIS is associated with fewer complications, less postoperative pain and less scarring after surgery [1]. The reduced pain and recovery time improves the patient’s quality of life and dramatically reduces the cost of surgery. 

Hospitals that widely adopt MIS techniques are estimated to save up to $340 million a year [2]. For these reasons, minimally invasive surgery has quickly become the standard for many surgical procedures, including cholecystectomies, which utilize MIS for 96% of operations [3]. However, despite the widespread adoption of MIS techniques, several limitations are still prevalent in the technology. For instance, current microtools with electrostatic or pneumatic actuation mechanisms require a connection to external control devices through tethers or tubes [4]. These tethers greatly restrict the maneuverability of the devices, which limits access to hard-to-reach regions of the body [5]. 

Another challenge associated with current MIS endoscopies is that clinicians cannot actively visualize and guide microtools to the target regions due to the lack of a method to control the positions of the devices [6]. Additionally, devices with electrostatic or pneumatic actuation mechanisms pose the risk of burns and burst failure. Some current microtools are actuated by a magnetic field; however, to actuate extremely small devices, a complex and impractically large magnetic field is required [4]. Furthermore, several actuation mechanisms of currently developed robotic and shape-memory grasping devices are difficult to miniaturize and are typically several centimeters large [5]. This relatively large size restricts access to narrow vessels and ducts within the body during a biopsy.

Thermomagnetic-responsive microgrippers have been developed to address the challenges currently associated with MIS technology. Microgrippers are star-shaped devices with multiple finger-like projections that can self-fold in response to changes in temperature. Compared to other mechanisms of actuation and folding, stimuli-responsive polymers offer several advantages that are valuable for minimally invasive surgeries and biopsies. A comparison between stimuli-responsive polymers and other actuation methods for microgrippers is shown in Table 1. 

Such benefits include their ability to self-fold without tethers and the ability to manufacture devices of extremely small size [4]. The sizes of these devices range between 300 μm to 1.5 mm, which is 10 to 100 times smaller than the current robotic grippers and biopsy forceps used for minimally invasive biopsy procedures [5]. Additionally, microgrippers are composed of thermal responsive polymer layers, such as poly(N-isopropylacrylamide-co-acrylic acid) (pNIPAM-AAc) or paraffin wax, which enable the devices to self-fold when exposed to changes in temperature. Furthermore, these devices are magnetic responsive due to the presence of magnetic materials or nanoparticles, such as iron oxide, embedded within the polymer layers or the metallic hinges of the devices. The thermomagnetic responsiveness of the devices allows surgeons to direct the microgrippers to the targeted region using magnetic fields and to propel the movement of the devices without the use of external tethers [5].

The small size and thermomagnetic responsiveness of microgrippers can improve minimally invasive surgeries and biopsies as shown in Figure 1. Tumor regions typically have an elevated temperature due to an increase in blood supply and inflammation [7]. When injected thermomagnetic microgrippers are exposed to these regions, the devices self-actuate due to the temperature change, allowing them to grip onto and excise cells. Surgeons can then utilize the magnetic response of the devices by applying a magnetic field to retrieve the cell samples for a minimally invasive biopsy. Several challenges exist for stimuli-responsive microgrippers. 

When compared to the actuation time of shape-memory actuators, which can self-actuate in seconds [8], thermal-responsive microgrippers have a slow response time as they tend to self-fold on the order of minutes [9]. Additionally, stimuli-responsive devices are dependent on the surrounding environment; however, this allows for autonomous actuation. Some further challenges that exist for thermomagnetic microgrippers include improving the visualization and navigation of microgrippers in vivo, improving the retrievability of the deployed grippers and optimizing the design of the devices to reduce the size, improve strength and increase precision. 

**Figure 1 molecules-27-05196-f001:**
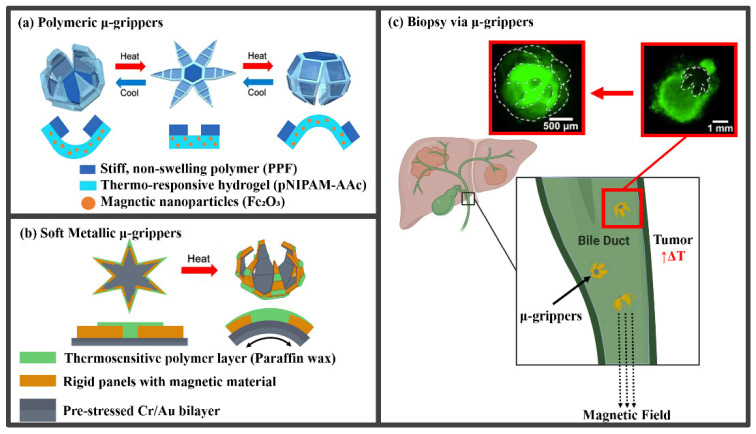
The use of polymeric and soft metallic thermomagnetic responsive microgrippers for minimally invasive biopsy. (**a**) Fully polymeric devices consist of a thermo-responsive poly(N-isopropylacrylamide-co-acrylic acid) (pNIPAM-AAc) hydrogel layer and stiff polypropylene fumarate (PPF) segments. The hydrogel layer swells or contracts when heated to above or below the transition temperature, causing the device to self-fold. Adapted with permission from Ref. [9]. Copyright 2015. (**b**) Soft metallic microgrippers consist of a pre-stressed bilayer of chromium and gold layered with rigid panels and thermosensitive paraffin wax hinges. The thermosensitive layer softens as the device reaches a temperature above normal body conditions, allowing the pre-stressed metallic layer to fold inward. Adapted with permission from Ref. [10]. Copyright 2014. (**c**) When exposed to the high-temperature regions of a tumor, the microgrippers can contract and grasp cells for a biopsy. Magnetic iron oxide particles in the porous hydrogel layer or magnetic materials in the rigid panels enable surgeons to use magnetic fields to control and retrieve the microgrippers with the excised tissue. Adapted with permission from Ref. [9]. Copyright 2015.

Stimuli-responsive polymers have the potential to serve as an excellent solution to limitations associated with current MIS technology. This review focuses on the mechanisms, fabrication, and applications of thermomagnetic-responsive microgrippers for tissue manipulation and excision during minimally invasive surgery. Additionally, a discussion on the current challenges and potential future improvements to thermomagnetic-responsive microgrippers is presented.

## 2. Mechanisms of Thermomagnetic Self-Actuating Microgrippers

Most of the advantages of using this technology stem from the ability of microgrippers to self-fold in response to temperature changes. In fully polymeric microgrippers, the thermal response comes from the pNIPAM-AAc hydrogel layer (Figure 2). Specifically, pNIPAM is a thermal responsive polymer with a lower critical solution temperature (LCST) in the range of 30 to 35 °C [11]. pNIPAM is copolymerized with acrylic acid (AAc), a hydrophilic monomer, to modify the LCST to 36 °C, which is closer to body temperature [9]. 

Furthermore, the addition of acrylic acid prevents complete polymer chain collapse, allowing for a more porous polymer network with greater swelling capacity [12]. pNIPAM-AAc is a random copolymer that features an amide linkage that can form hydrogen bonds with water molecules or amides present in other pNIPAM-AAc chains, enabling the material to change solubility. The configuration of the multi-fingered microdevices with thermal-responsive pNIPAM hinges changes as the solubility of the polymer varies with temperature. 

At temperatures below 36 °C, the polymer acts hydrophilic as the hydrogen bonding between water and the polymer chains is energetically favorable. This results in an increase in water–polymer interactions and a decrease in polymer–polymer interactions. Therefore, the pNIPAM hydrogel layer absorbs water, resulting in swelling. The increase in the volume of the hydrogel layer causes the microgripper to bend and close its finger-like appendages [9]. At temperatures above the LCST, including the average body temperature of 37 °C, the pNIPAM becomes hydrophobic. Polymer–polymer interactions increase with temperature, resulting in a decrease in water–polymer interactions [13]. The decrease in hydrogen bonding between water and polymers makes it thermodynamically unfavorable for water to remain inside the polymer network. 

As the hydrogel layer shrinks with water loss, the fingers of the microgripper self-fold to close in the opposite direction [9]. The mechanism of opening and closing thermal-responsive polymeric microgrippers with temperature changes is displayed in Figure 3.

For soft metallic microgrippers, the mechanism of self-folding is also in response to temperature changes. However, rather than temperature inducing a change in the solubility of the polymer, temperature causes the polymer to deform. Typically, soft robotic microgrippers have a layer of thermosensitive paraffin wax that holds the pre-stressed metallic bilayer flat at low temperatures due to the rigidity of the wax. The chemical structure of paraffin wax is shown in Figure 4. Only weak van der Waals bonds form between paraffin molecules due to their long carbon-chain backbones [15]. As temperature increases, these weak bonds are broken, and the thermo-responsive paraffin wax layer softens and triggers the self-folding of pre-stressed metallic hinges [10]. 

This self-actuation is induced by a change in the Young’s modulus of paraffin wax. Young’s modulus is determined by the slope of the stress–strain curve. As shown in Figure 5, the Young’s modulus, which is the measure of a material’s stiffness, decreases significantly from 6.44 MPa at 26 °C to 0.37 MPa at 37 °C [16]. This decrease indicates that the paraffin wax layer becomes soft and easy to bend. The softened paraffin wax layer exerts less force on the metallic bilayer of soft robotic microgrippers, allowing the pre-stressed hinges to release the stored stress, causing the fingers of the gripper to fold and close [17].

Another key feature of thermomagnetic microgrippers is their magnetic responsiveness, which allows for the user to have control of the position of the injected devices. In thermal-responsive polymeric microgrippers, the magnetic responsiveness is synthesized by incorporating magnetic nanoparticles into the device’s hydrogel network. Typically, iron oxide (Fe₂O₃) is incorporated into the pNIPAM-AAc layer of polymeric microgrippers [9]. These nanoparticles are magnetic, so their dispersion throughout the polymer layer allows the devices to be guided with a magnetic field.

In soft robotic metallic microgrippers, the metallic bilayer hinges contain magnetic materials that enable the device’s position to be controlled via magnetic fields [17]. As shown in Figure 6A, the ferromagnetic material is embedded within the rigid gold panels of the gold-chromium bilayer. These magnetic materials are typically iron, nickel, or cobalt due to their high magnetic dipole moment. A comparison of the magnetic dipole moments of microgrippers made with various magnetic metals is shown in Figure 6B. Microgrippers with cobalt embedded within them had the greatest dipole moment of 11 μA $m^{2}$; however, iron is most often used due to its increased biocompatibility [17]. A high magnetic dipole indicates that the devices have a high response to an applied magnetic field.

## 3. Fabrication and Design of Thermomagnetic-Responsive Microgrippers

Both types of microgrippers, polymeric and soft metallic, are commonly modeled after a hand, with several flexible fingers that form a star-like shape [5]. The multi-fingered design of thermomagnetic-microgrippers allows the devices to grip onto tissue and other small molecules. Typically, these devices feature four to six fingerlike projections to ensure that the devices can hold their grasp when being directed under high magnetic fields [18]. Furthermore, for the devices to be capable of excising cells for biopsies, the microgrippers must be large enough to encapsulate cells within the device. Cells range in size from 10 to 100 µm, making the ideal size for microgrippers to be between 10 µm to 1 mm. Due to the fabrication methods, microgrippers range in size from 300 μm to 1.5 mm, making them large enough to capture cells within the fingers of the device but small enough to be easily delivered [5].

Polymeric microdevices are designed to have thermal-responsive hydrogel hinges paired with stiff polymer segments. The hydrogel layer is typically made up of pNIPAM-AAc due to its thermal responsiveness, swelling ability and low shear modulus of 162 kPa [9]. Polypropylene fumarate is often used to make up the stiff polymer segments due to its nonswelling behavior and high shear modulus of 16 MPa [9]. SU-8 is an alternative polymer that can make up the stiff segments of microgrippers [19]. SU-8 is a photopatternable epoxy resin with a high elastic modulus of 3.6 GPa and excellent chemical and thermal stability [20]. 

The PPF or SU-8 segments add stiffness to the microgripper and increase its gripping strength. The stiff segments restrict the swelling behavior of the pNIPAM-AAc layer, causing the fingers of the microgripper to bend in an origami-like manner [9]. The thickness of the two polymer layers has a large effect on the folding ability of the device. An increase in thickness in the PPF layer results in a lower degree of folding as the PPF increases the rigidity of the device [9]. A lower degree of folding also occurs when the pNIPAM-AAc layer is thin. 

A thin hydrogel layer is unable to exert enough force onto the stiff PPF segments to elicit a bending response [9]. However, low folding still occurs when the pNIPAM-AAc layer is thick as the thick hydrogel layer experiences a negligible force from the PPF segments [9]. For proper folding behavior, the optimum thickness of the PPF segments is 10 μm, while the ideal pNIPAM-AAc layer thickness is 45 μm [9]. Figure 7 displays the effect that PPF and pNIPAM-AAc thickness has on the folding ability of polymeric microgrippers. The diameter reduction ratio (D/D₀) refers to the ratio of the folded diameter to the unfolded diameter. A smaller diameter reduction ratio corresponds to a greater degree of folding.

Soft robotic microgrippers are composed of pre-stressed metallic bilayer hinges that are flattened with a thermosensitive polymer layer [10]. Paraffin wax is most often used for the thermosensitive polymer trigger layer. This material was chosen due to its biocompatibility and chemical and biological inertness [16]. Furthermore, the phase-transition temperature of paraffin wax is near body temperature [16]. The thickness of the paraffin wax layer plays a large role in the bending ability of the microgripper. 

As the thickness increases, the degree of gripper folding decreases due to an increase in rigidity [13]. The relationship between the fold angle of the microgripper fingers and the thickness of the paraffin wax layer is shown in Figure 8A. The optimal thickness of the paraffin wax layer is between 0.5 and 1 μm, as indicated by the shaded region of Figure 8A. In this thickness range, the microgripper has a low fold angle at the low temperature and a high fold angle at the 37 °C trigger temperature [13]. Figure 8B shows some finite element models of microgripper folding at low and high temperatures when the paraffin wax has a thickness of 0.6 and 0.9 μm [16].

Polymeric thermomagnetic-responsive microgrippers are fabricated through serial photolithography [9]. The fabrication process of polymeric pNIPAM-AAc and PPF microgrippers is displayed in Figure 9A. First, a layer of PPF is deposited onto a silicon wafer through spin coating and cross-linked by applying UV light through a dark field mask. Next, pNIPAM-AAc mixed with 5% weight concentration iron oxide is poured on top of the PPF layer. This layer is then also cross-linked by exposing the polymer layer to UV light through a dark field mask. Uncrosslinked polymers are then removed by submerging the silicon wafer in alcohol to reveal the star-shaped polymeric microgrippers [9].

For soft metallic microgrippers, photolithography with electrodeposition is utilized to fabricate the metallic bilayer, then the thermo-sensitive polymer is spin-coated to the surface [6]. Figure 9B displays the fabrication process of soft robotic microgrippers. First, the stressed metallic bilayer and the embedded magnetic material are created through UV photolithography [6]. The metal layers are then electrodeposited onto a mold where molten paraffin wax is spin coated onto the metal components. Once cooled, the microgripper is encapsulated within a rigid paraffin wax shell [17].

**Figure 9 molecules-27-05196-f009:**
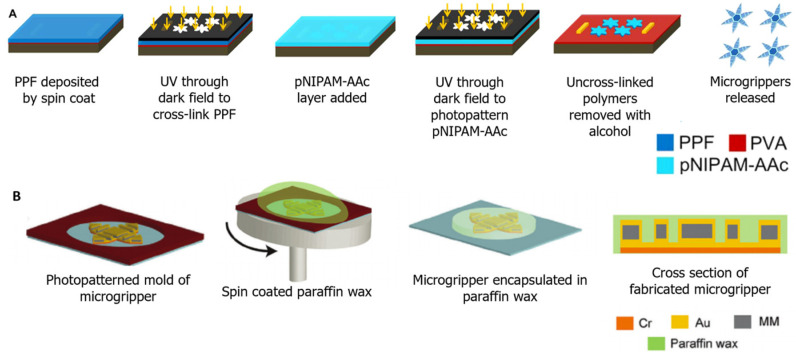
Fabrication of thermomagnetic microgrippers. (**A**) The fabrication process of polymeric microgrippers through serial photolithography. Adapted with permission from Ref. [9]. Copyright 2015. (**B**) The fabrication process of soft metallic microgrippers. Metallic layers are created through UV photolithography then electrodeposited as a mold. Paraffin wax is then spin-coated over the mold to encapsulate the microgripper. Adapted with permission from Ref. [17]. Copyright 2021.

## 4. Applications of Thermomagnetic-Responsive Microgrippers in Minimally Invasive Surgery

Thermomagnetic-responsive microgrippers have to capability to be used to improve minimally invasive biopsies. When exposed to an elevated temperature in the presence of a tumor, the fingers of the microgrippers can self-fold to grasp onto the surrounding tissue and excise cells for a minimally invasive biopsy. Furthermore, the magnetic responsiveness of the microgrippers allows surgeons to easily retrieve the deployed microgrippers and the tissue captured by the devices to perform DNA disease-diagnostic analysis. In vitro testing of polymeric microgrippers composed of PPF and pNIPAM-AAc verified the thermal responsiveness of the devices. 

Microgrippers submerged in solution were able to fold in both directions in response to temperature changes as shown in Figure 10A. Above 36 °C, the gripper is closed with the PPF segments on the external surface. The hydrogel layer appears opaque, indicating that the polymer network is collapsed and that the pNIPAM-AAc is hydrophobic [9]. It was also verified that, below 36 °C, the microgripper closes in the opposite direction with the PPF segments on the inside surface of the gripper. 

The hydrogel layer appears clear at this temperature, indicating that the polymer network is swollen with water and hydrophilic. This self-folding and unfolding behavior was observed for over 50 cycles between 28 °C and 43 °C confirming that the self-actuation mechanism is reversible. These results confirmed that pNIPAM-AAc microgrippers self-fold at temperatures near-physiological conditions and that they are capable of self-actuating multiple times, which could be useful for storage prior to surgery.

The gripping strength of polymeric microdevices was also verified in vitro to be strong enough to extract cells from tissue. As seen in Figure 10B, pNIPAM-AAc and PPF microgrippers were able to extract fibroblast cells from a cell clump when pipetted from a cold storage solution to a 37 °C environment [9]. These results validated the ability of the microgrippers to excise cells from tissue without the use of tethers, indicating potential usefulness for cell collection during minimally invasive biopsies.

In vitro experiments also confirmed that the position of polymeric microgrippers doped with iron oxide can be manipulated through the application of a magnetic field. Figure 10C shows images of a pNIPAM-AAc and PPF microgripper with 5% *w*/*w* iron oxide. The position of the device was able to be controlled with a magnetic probe, indicating the potential for the microgrippers to be controlled via magnetic fields in clinical settings [9]. In vitro testing verified that microgrippers controlled via a closed-loop system were also capable of manuevering around obstacles while performing pick-and-place procedures. 

In this experiment, microgrippers composed of pNIPAM-AAc and SU-8 segments were controlled by a motion planning algorithm via electromagnets. The microgrippers were able to avoid both static and moving virtual obstacles while moving a piece of egg yolk. The experiment was performed on a base of porcine tissue and egg yolk was used to mimic the fat and cholesterol levels of materials present during MIS procedures. The average error of the egg yolk drop-off was 0.62 ± 0.22 mm, and the grippers containing biological material were able to move at an average velocity of 1.81 mm/s, verifying that thermomagnetic-responsive microgrippers are capable of manipulating biological materials in complex environments [19].

The targeting precision of deployed microgrippers was also investigated in vitro. Sixty-six soft robotic microgrippers were delivered into a water bath through an endoscope [10]. The distance between the delivery point and the final positions of the deployed microgrippers was measured. The results of this experiment are displayed in Figure 11. An average of 40% of the microgrippers stayed within 10 mm of the delivery point, which indicated that the microgrippers are precise and will stay in the targeted region when deployed [10]. 

The ability of thermomagnetic-responsive microgrippers to autonomously recognize and sort materials was also verified in vitro. Closed-loop controlled microgrippers were able to recognize colored polyester beads and move each one to the area marked with the corresponding color. The spherical beads were 0.5 mm in diameter and weighed around 0.6 mg. The average error of the object drop-off was 0.85 ± 0.41 mm, which fell into the range of the drop-off location boundaries. This demonstrated that microgrippers can also autonomously recognize and manipulate micro-scale objects with high precision [19].

The feasibility of utilizing thermomagnetic-responsive microgrippers to perform tissue excision in reach hard-to-reach areas of the body was confirmed through an ex vivo study performed on a porcine model. An endoscope was inserted in the mouth of a pig and advanced to the duodenum. Soft robotic microgrippers (1560) were then deployed into the common bile duct of the pig. A catheter with a magnetic tip was used to collect the deployed microgrippers 10 min after insertion to allow the devices to close in response to the body temperature. On average, 95% of the deployed microgrippers were successfully retrieved [5]. 

These retrieved microgrippers successfully extracted tissue from the biliary duct. The volume of tissue collected varied greatly with the number of microgrippers deployed and retrieved. Larger tissue samples were able to be analyzed via cytological analysis. DNA from small tissue samples was able to be successfully extracted and amplified through polymerase chain reaction (PCR) for analysis of disease-diagnostic measures. This experiment verified that soft robotic microgrippers can retrieve enough tissue to identify disease-diagnostic markers, which is extremely useful in minimally invasive biopsies [5].

## 5. Outlook and Challenges Associated with Thermomagnetic-Responsive Self-Folding Microgrippers

Several studies verified the ability of both polymeric and soft robotic microgrippers to self-fold in response to temperature change, be guided with magnetic fields and excise enough tissue to analyze disease markers. However, several challenges must be addressed before this technology can be adopted in clinical environments. One major challenge that will become the focus of future research is developing an improved method of monitoring the positions of the microgrippers. Surgeons currently have trouble visualizing the microgrippers once inserted. 

The compatibility of ultrasound, magnetic resonance, or near-infrared imaging with monitoring microgripper location can be investigated in the future to establish a clinically available method of viewing microgripper position [4]. Additionally, improvements to the external control method of directing microgrippers could be investigated. Due to the metallic properties of magnetic-responsive microgrippers, three-dimensional magnetic manipulation systems, such as magnetic-resonance-imaging-based guidance methods, could be utilized in the future to steer and control the position of microgrippers more efficiently [5]. Alternatively, combining polymeric microgrippers with bacteria could utilize bacteria chemotaxis as a locomotion method to deliver microgrippers to specific locations in vivo without the use of magnetic fields [6].

Another major challenge is preventing microgrippers from being left behind in the body. Minimizing the number of devices left behind improves the retrieval of excised samples and the biocompatibility of the devices. Despite the 95% retrieval rate in the ex vivo study discussed, other studies have had significant problems with retrieving deployed microgrippers; therefore, a more consistent retrieval method needs to be developed. 

One potential solution to the low retrieval yield of microgrippers is to apply a stronger magnetic field to externally control the devices [10]. Another potential direction for future research is to fabricate thermogrippers from biodegradable materials to reduce the long-term impact and increase the biocompatibility. Thermal-responsive biodegradable hydrogel networks, such as poly(oligoethylene glycol methyl ether methacrylate (Mn = 500)-bis(2-methacryloyl)oxyethyl disulfide) (P(OEGMA-DSDMA)), are being explored as a possible alternative to pNIPAM-AAc hydrogel layers [21].

While the finger-like design has been verified to be capable of extracting tissue for DNA analysis, design optimization of microgrippers is a major goal for continuous improvement to the technology. Several modifications to the shape of microgrippers are being explored to reduce fabrication cost, improve strength, and increase degrees of freedom. One design option being investigated is a simple bilayer microgripper. This design reduces the fabrication steps and the costs associated with hinged microgrippers. 

Researchers have used finite element analysis to demonstrate the gripping and transport capabilities of this design concept, which could further be applied to thermomagnetic responsive microgrippers to simply the manufacturing process [22]. Another possible design that can be applied to microgrippers is a smart microgel on a photopatterned base. A microgel made of pNIPAM and poly(ethylene glycol) diacrylate (PEGDA) enables the gripper to conform itself around an object and grip due to the shape-memory properties of the microgel [23]. 

This design offers infinite degrees of freedom and can exert gripping forces greater than 400 μN. Furthermore, the capability for tetherless guidance has been verified as microgrippers incorporated with Nickle particles were able to grasp objects and navigate through a maze using magnetic forces [23]. Therefore, microgels could pose as intriguing solutions to improving the fabrication process and increasing the conformability of microgrippers for minimally invasive operations in the future.

Future research will likely also explore methods to further reduce the size of thermomagnetic microgrippers. The fabrication methods of microgrippers are well-developed across numerous materials and scales of size; therefore, it is reasonable to consider further miniaturization of microdevices in the future [5]. By reducing the size of the microgrippers, the devices will become more capable of maneuvering into narrow vessels of the body, expanding the applicability of the devices to vascular procedures [4]. Alternative fabrication methods, including 3D printing, could be applied to the manufacturing of thermomagnetic microgrippers. 

Thermal-responsive pNIPAM hydrogel actuators and pNIPAM bilayers have been previously fabricated using 3D printing technology [24,25]. Furthermore, microscale continuous optical printing (μCOP) has been utilized to ED print microscale fish composed of a PEDGA-based hydrogel. Iron oxide nanoparticles were also able to be printed into the hydrogel matrix of the microfish through the μCOP system [25]. This 3D printing technology shows potential for simplifying the fabrication process of thermomagnetic-responsive microgrippers in the future.

Improving the response time and precision of thermomagnetic-responsive microgrippers is another major challenge. Temperature-responsive microgrippers have the slowest response time and, therefore, poor precision. By utilizing different actuation stimuli, the response rate and precision of the devices could be increased [6]. Laser activated soft metallic microgrippers with a Gold metallic layer could pose as a future solution for faster and more controlled thermal actuation. Gold has a high light absorption and microgrippers composed of Gold have been found to actuate in 60 ms [26]. Therefore, the integration of Gold and laser light activation have the potential to greatly increase the response time of thermomagnetic microgrippers. 

Furthermore, the precision of the microgrippers could be increased by fabricating the individual fingers out of various stimuli-responsive materials. This would allow for independent control over the fingers of the microgrippers and increased precision [6]. Control over the individual fingers of microgrippers and the utilization of alternative materials could lead to future implementation of thermogrippers for more advanced minimally invasive surgical techniques, such as cutting or sewing [5]. 

Additionally, incorporating microelectromechanical systems (MEMS) technology into hydrogel based microgrippers has the potential to increase dexterity while maintaining the mechanical strength of the devices. Preliminary work has shown that MEMS hydrogel actuators can contract in milliseconds while maintaining a high gripping force [27]. The increase in dexterity and actuation speed associated with MEMS technology could allow for better micromanipulation during minimally invasive surgery with thermomagnetic-responsive microgrippers.

## 6. Summary

In conclusion, thermomagnetic-responsive microgrippers have the potential to revolutionize minimally invasive surgical techniques. With the ability to self-fold in response to temperature changes and the ability to be guided via a magnetic field, thermomagnetic-responsive microdevices eliminate the need for external control tethers, which greatly increases the accessibility and maneuverability of the devices. Microgrippers are 10- to 100- times smaller than the current biopsy forceps or robotic grippers used for minimally invasive biopsies. 

This small size allows the devices to access extremely hard-to-reach areas of the body and greatly reduces the pain and discomfort associated with surgery, leading to faster recovery times and reduced surgery costs. Polymeric microgrippers doped with iron oxide nanoparticles self-actuate in response to temperature changes due to changes in the solubility of the polymer network. On the other hand, soft robotic microdevices self-fold due to a change in the Young’s modulus of the polymer trigger layer. The sharp tips of the star-shaped design allow these devices to dig into surrounding tissue and excise cells. 

The thermal responsiveness of both fully polymeric and soft robotic microgrippers enables the devices to self-fold when exposed to high-temperature tumor regions. The magnetic responsiveness allows surgeons to easily control the position of the microgrippers and retrieve tissue samples containing DNA for minimally invasive biopsies. Future research around thermomagnetic-responsive microgrippers will focus on improving device tracking methods, reducing the number of devices left in the body and expanding this technology to other surgical techniques.

## Figures and Tables

**Figure 2 molecules-27-05196-f002:**
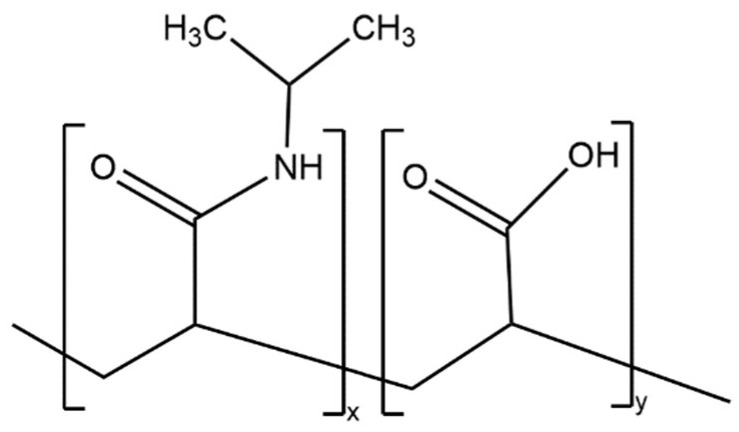
Chemical structure of poly(N-isopropylacrylamide-co-acrylic acid) (pNIPAM-AAc). This random copolymer has an LCST of 36 °C.

**Figure 3 molecules-27-05196-f003:**
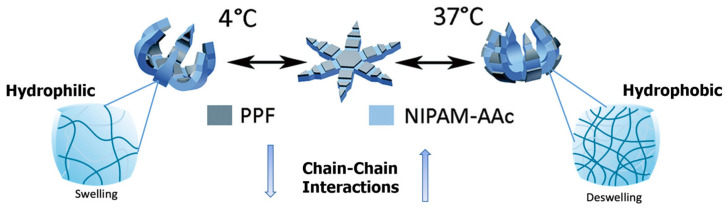
Mechanism of pNIPAM and PPF thermoresponsive microgrippers. At temperatures below the LCST of 36 °C, the pNIPAM-AAc polymer acts hydrophilic and swells, causing the microgripper to self-fold. Above 36 °C, pNIPAM-AAc acts hydrophobic, and the hydrogel network shrinks, causing the microgripper to fold in the opposite direction. Adapted with permission from Ref. [13] Copyright 2020 and [14]. Copyright 2014.

**Figure 4 molecules-27-05196-f004:**
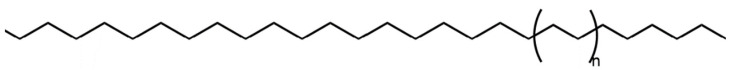
Chemical structure of paraffin wax. The molecule is a long chain of carbons and hydrogens. The chemical formula of paraffin wax is C_*n* H_((2*n* + 2)), where *n* is between 20 to 40.

**Figure 5 molecules-27-05196-f005:**
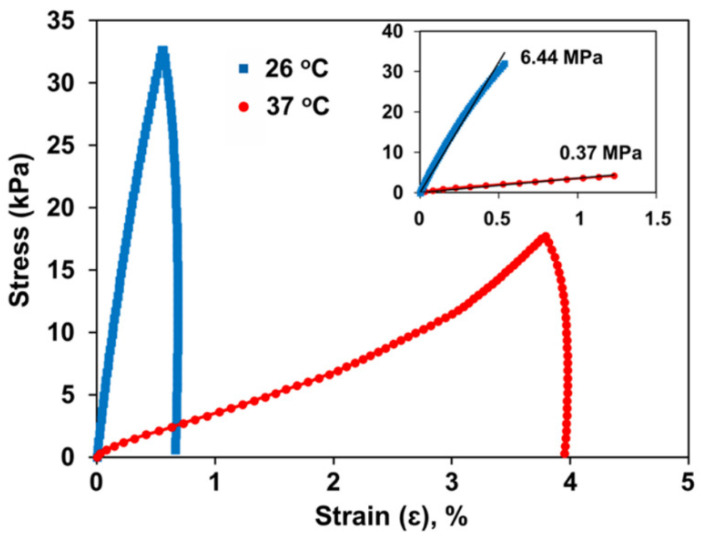
Stress–strain curve for paraffin wax at various temperatures. The Young’s modulus of the material decreases significantly with increasing temperature. This decrease in paraffin wax stiffness with increased temperature is the mechanism behind the self-folding ability of soft metallic microgrippers. Reproduced with permission from Ref. [16]. Copyright 2020.

**Figure 6 molecules-27-05196-f006:**
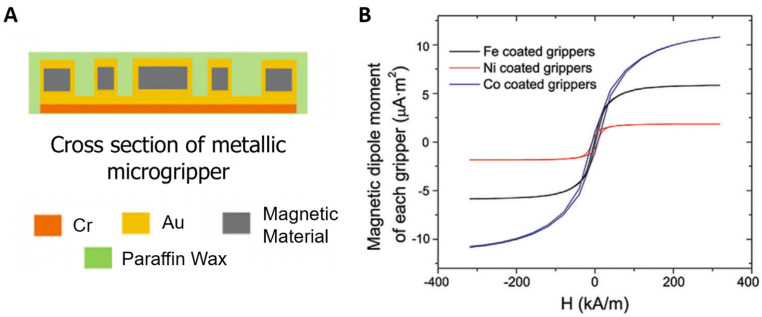
Magnetic response in metallic thermogrippers. (**A**) As seen in the cross-section of a metallic thermoresponsive microgripper, magnetic material is embedded within the Cr-Au bilayer to elicit a magnetic response. (**B**) The magnetic dipole moments of soft robotic microgrippers with various magnetic metals layers were compared. Microgrippers with cobalt had the greatest magnetic dipole moment of 11 μA $m^{2}$. Adapted with permission from Ref. [17]. Copyright 2021.

**Figure 7 molecules-27-05196-f007:**
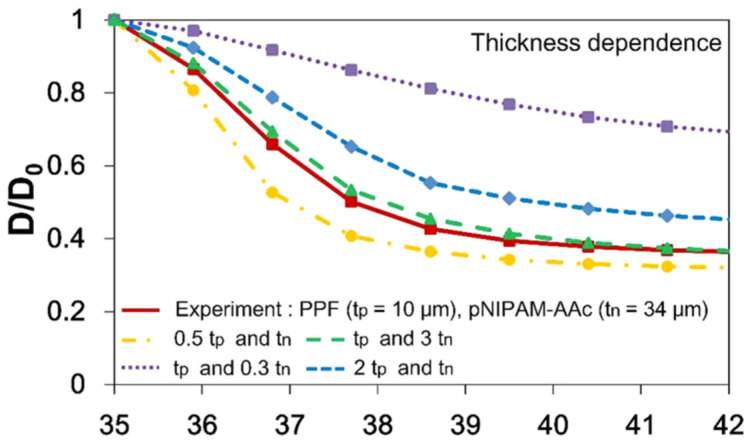
PPF and pNIPAM-AAc thickness and folding behavior. The thickness of the PPF and pNIPAM-AAc layers has a large effect on the folding behavior of polymeric devices. Reproduced with permission from Ref. [9]. Copyright 2015.

**Figure 8 molecules-27-05196-f008:**
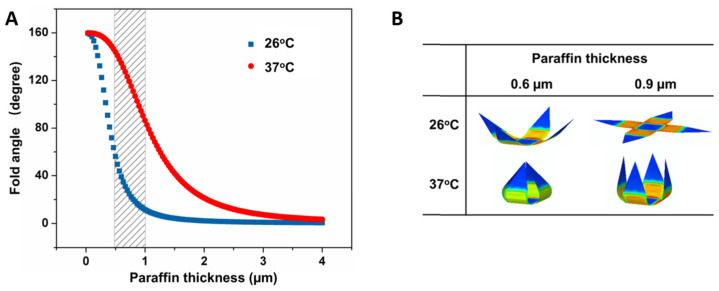
Effect of paraffin wax thickness on folding behavior. (**A**) As the thickness of the paraffin wax layer increases, the fold angle of the microgripper fingers decreases due to an increase in device rigidity. (**B**) Finite element modeling of microgrippers within the optimal thickness range shows the folding behavior of the devices at 26 and 37 °C. Reproduced with permission from Ref. [16]. Copyright 2020.

**Figure 10 molecules-27-05196-f010:**
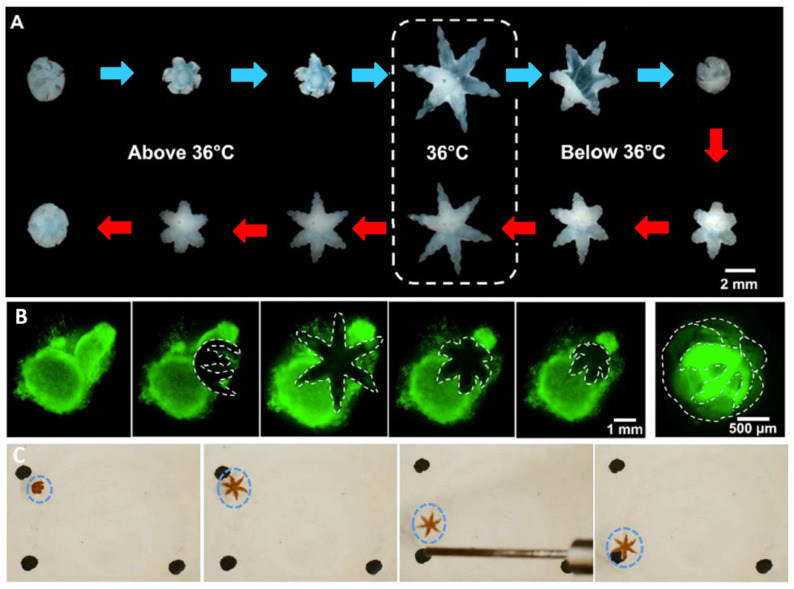
In vitro verification of thermomagnetic-responsive microgrippers. (**A**) PPF and pNIPAM-AAc microgrippers demonstrated the ability to self-fold and unfold with temperature changes for over 50 cycles. (**B**) Microgrippers demonstrated the ability to excise and capture fibroblast cells from a cell clump. Dashed lines indicate the boundary of the clear microgripper. (**C**) Polymeric microgrippers doped with iron oxide were able to be guided from one marker to another with a magnetic probe. Adapted with permission from Ref. [9]. Copyright 2015.

**Figure 11 molecules-27-05196-f011:**
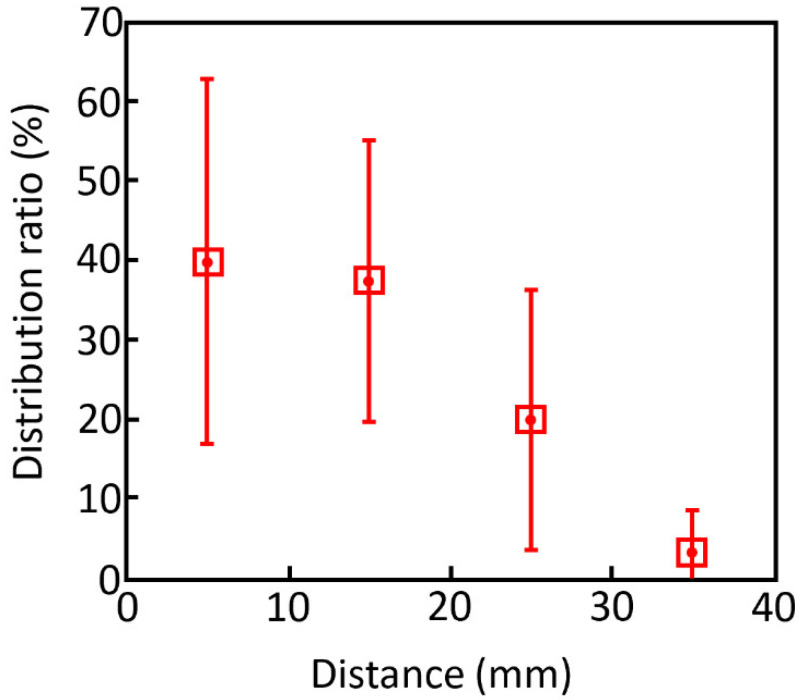
Distribution ratio of deployed microgrippers over distance from the targeted delivery point. Reproduced with permission from Ref. [10]. Copyright 2014.

**Table 1 molecules-27-05196-t001:** Comparison of Common Actuation Mechanisms for Microgripping Technology. Reprinted with Permission from Ref. [4]. Copyright 2017.

Actuation Mechanism	Advantages	Drawbacks
**Electrostatic/ionic**	Precise control of local motion Reproducible response Reversible response	Needs a tether Often require high voltage Concerns with electrical burns Challenging to miniaturize Slow response Short lifetime
**Pneumatic/fluidic**	Precise control of local motion Safe for in vivo use Insensitive to the environment Complex actuation possible	Needs tubing Needs external gas, fluid, compressor Challenging to miniaturize Challenges with fittings, burst failure
**Magnetic**	Untethered actuation Can be miniaturized Independent of composition of the environment	Actuation setup can be very complicated Large impractical 3D magnetic fields and gradients may be required to actuate very small structures in humans
**Shape memory materials**	Untethered actuation Safe for in vivo use Fast actuation Independent of composition of the environment	Irreversible actuation In most cases, temperature is the only stimulus, which can limit applicability Pre-deformation often required Challenging to miniaturize
**Stimuli responsive polymers**	Untethered actuation Extreme miniaturization possible Autonomous Ease of manufacturability; can be 3D printed	Typically slow Dependent on the composition of the environment

## Data Availability

Not applicable.

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
