# Peer review of "Thermomagnetic-Responsive Self-Folding Microgrippers for Improving Minimally Invasive Surgical Techniques and Biopsies"

_molecules, 2022, doi:10.3390/molecules27165196_

Round 1

Reviewer 1 Report

In this review, the authors showed thermomagnetic-responsive self-folding microgrippers for improving minimally surgical techniques and biopsies. This review gave detailed information about thermomagnetic-responsive microgrippers with mechanisms, fabrication, design, and applications in minimally invasive surgery. Then the outlook and challenges associated with thermomagnetic-responsive self-folding microgrippers also were reviewed. Thus, I recommend it be accepted.

Author Response

The authors would like to thank the reviewer for their thoughtful comments. 

Reviewer 2 Report

The manuscript is generally very interesting. However, it needs some revisions before it can be accepted for publication.

Please reposition the title of the figures that must be separated from the normal text. For the figures caption, only the number of the figure must be in bold, the rest is regular. The same for the tables.

Please write the titles for Figure 8 and Figure 10.

Please leave a space between the above text and figure 10.

If possible, try to eliminate rows 373 and 392.

Author Response

The authors would like to thank the reviewer for their thoughtful and specific comments. We feel the manuscript has been strengthened by inclusion of the feedback. The responses to these comments are below and also reflected in the revised text of the manuscript.

Comment 1: Please reposition the title of the figures that must be separated from the normal text. For the figures caption, only the number of the figure must be in bold, the rest is regular. The same for the tables.

These changes have been made in the updated manuscript as requested.

Comment 2: Please write the titles for Figure 8 and Figure 10.

Titles have been added for Figures 8 and 10.

Comment 3: Please leave a space between the above text and figure 10.

A space has been added as suggested.

Reviewer 3 Report

1. The introduction substantiates the prospects for the use of thermomagnetic-sensitive microgrippers in surgery. In addition, the review highlights the issues of their manufacture and problems with their use. This should be indicated in the introduction to the review.

2. The research results presented in the review indicate the unique properties of polymeric materials based on poly(N-isopropylacrylamide-co-acrylic acid) and polypropylene fumarate. I would like to see in the article the conditions for obtaining and the most important characteristics of these polymers.

3. Signatures under the drawings are very voluminous. Part of the description from the captions to the figures can be transferred to the text.

4. The review summarizes the results of research on the specified problem. The number of publications containing information on this problem over the past 5 years is much more than is given in the article. The list of references should be supplemented.

5. The list of references should include studies conducted on the problem under consideration over the past 5 years. This indicates the urgency of the problem. Almost half of the list of references does not comply with this rule.

Author Response

The authors would like to thank the reviewer for their thoughtful and specific comments. We feel the manuscript has been strengthened by inclusion of the feedback. The responses to these comments are below and also reflected in the revised text of the manuscript.

Comment 1: The introduction substantiates the prospects for the use of thermomagnetic-sensitive microgrippers in surgery. In addition, the review highlights the issues of their manufacture and problems with their use. This should be indicated in the introduction to the review.

The introduction has been edited to include manufacture and use issues. 

Comment 2: The research results presented in the review indicate the unique properties of polymeric materials based on poly(N-isopropylacrylamide-co-acrylic acid) and polypropylene fumarate. I would like to see in the article the conditions for obtaining and the most important characteristics of these polymers.

Important processing and characteristics points have been added throughout the revised manuscript where applicable.

Comment 3: Signatures under the drawings are very voluminous. Part of the description from the captions to the figures can be transferred to the text.

Portions of the figure captions have been incorporated into the text of the revised manuscript.

Comment 4: The review summarizes the results of research on the specified problem. The number of publications containing information on this problem over the past 5 years is much more than is given in the article. The list of references should be supplemented.

The list of reference has been expanded to include additional relevant publications.

Comment 5: The list of references should include studies conducted on the problem under consideration over the past 5 years. This indicates the urgency of the problem. Almost half of the list of references does not comply with this rule.

References added include most recent (last 5 years) that would more adequately  convey the urgency of the problem.